The effect of enhanced variability after performance stabilization through constant practice

Ruano Carmen
Caballero Carla
http://orcid.org/0000-0002-2520-892X Moreno Francisco J. fmoreno@umh.es
Sports Research Centre, Universidad Miguel Hernández de Elche , Elche, Alicante , Spain
Lavender Andrew
Electronic publication date: 2022 Sep 16
Publication date: 2022
Volume: 10
Electronic Location ID: e13733
Received 2021 Aug 24; Accepted 2022 Jun 24
Copyright: © 2022 Ruano et al.
Copyright year: 2022
Copyright holder: Ruano et al.
License: This is an open access article distributed under the terms of the Creative Commons Attribution License, which permits unrestricted use, distribution, reproduction and adaptation in any medium and for any purpose provided that it is properly attributed. For attribution, the original author(s), title, publication source (PeerJ) and either DOI or URL of the article must be cited.
License URL: https://creativecommons.org/licenses/by/4.0/

Keywords: Motor learning, Constant practice, Variable practice, Adaptive learning, Inherent variability

Funding: Economy, Industry and Competitiveness Ministry of Spain DEP2016-79395-P This work was supported by the Economy, Industry and Competitiveness Ministry of Spain, under Grant cod. DEP2016-79395-P. The funders had no role in study design, data collection and analysis, decision to publish, or preparation of the manuscript.

==============================
There is a controversy about the benefits of variable practice on motor learning. This study aimed to analyze the effect of variable practice after the learner stabilized their performance. Thirty-two healthy adults performed a computer-simulated discrete accuracy task in which they had to release a virtual ball to try to hit a target. After a pre-test, the participants were distributed into three training groups: constant practice, variable practice and a group that started practicing in constant conditions, and when their performance stabilized, changed to variable practice. The participants performed 44 sets of 10 repetitions of the task. A post-test and two retention tests were carried out. Variable practice did not show a significant improvement compared to constant practice. Variable practice did not show higher benefits even when performance was stabilized through constant practice, but its effects seem to be modulated by the initial variability of the learners.

Introduction

Variability is an inherent feature of motor behaviour, which can be observed in all living beings and it seems to be linked to the ability to adapt to the environment. In reinforcement learning, motor variability would allow the learners to explore the situations they are faced with, by adapting their outputs through movement variations, trying to find the best motor solution to every situation (Wu et al., 2014). This has also been explained through the dynamic systems theory (DST), under the idea that the fluctuations in motor behavior can improve the range of solutions that a person could use to fulfill the demands of a specific task (Davids et al., 2003). One of the most studied topics to facilitate motor learning has been the role of induced enhanced variability, which is known as variable practice. The formulation of variable practice was originally based on the idea of acquiring a generalized motor program in the schema theory (Schmidt, 1975). That perspective argued that variable practice facilitates the development of a more flexible schema in unpredictable environments, or open skills. More recent studies, under the DST, have argued that this kind of practice facilitates not only a better performance in open environments but it also facilitates motor learning processes in closed skills (Savelsbergh et al., 2010; Davids et al., 2003; Davids, Bennett & Newell, 2006; Stergiou, Harbourne & Cavanaugh, 2006; Thelen, 1995).

Previous studies have addressed how variable practice, understood as the modifications of the standard movement or task patterns of an ability had a positive effect in the learning of fundamental movement skills (Caramiaux et al., 2018; Dhawale, Smith & Ölveczky, 2017), and in individual and collective sport skills (Menayo et al., 2012; Urbán, Hernández-Davó & Moreno, 2012; Williams & Hodges, 2005). However, other studies have not found those positive results for the variable practice vs. constant practice, obtaining better results with this last one (Chu, Sternad & Sanger, 2013; Garcia-Herrero et al., 2016; Wulf, Lee & Schmidt, 1994). These results lay out a controversy in the application of this training method.

An important factor that needs to be taken into consideration when interpreting the results of variability training is the magnitude of the external variability load applied. According to McEwen & Lasley (2002), the body’s response to a physical training load has an inverted “U” shape, in which very low load levels will not be enough to cause adaptations, the same as very high levels could be detrimental, causing non-desired adaptations. Regarding motor learning, the challenge point hypothesis (Guadagnoli & Lee, 2004) suggested that learning will be reduced in the presence of too little or too much information. If a task provides little information (e.g. a slight difficulty for an expert performer), the learner will learn very slowly. On the other side, a very difficult, or a very rich informative task for a novel performer will also diminish the learning rate. From this hypothesis, the optimum amount of practice load would be a function of the skill level of the individual (functional task difficulty) and the difficulty of the to-be-learned task (nominal task difficulty). This idea was taken by Moreno & Ordoño (2015), who extrapolated it to variable practice load. Thus, there are studies that propose a moderate variability magnitude as the optimal level to obtain positive results (Caballero et al., 2017; Harbourne & Stergiou, 2009; Ranganathan & Newell, 2010). In this sense, some authors have pointed out that practice individualization based on the characteristics of the individual is a key issue that should be taken into consideration when training variability is applied (Caballero et al., 2017). In a study carried out with handball players by Garcia-Herrero et al. (2016), it was confirmed that the most experienced players reported greater improvements with higher levels of practice variability, while those who were beginners needed lower levels, or even the absence of extrinsic variability. That would be due to the fact that their own exploratory behaviour offered enough internal variability, in line with the concept that was introduced. Thus, inducing a high level of external variability in individuals with a low level of dexterity could delay the search of motor solutions when facing the learning of a motor task (Wulf, Lee & Schmidt, 1994). This would be in accordance with the idea of how the initial level of the learner could be a factor to be taken into consideration to apply induced variability, as beginners usually show larger motor variability than experienced learners (Leversen, Haga & Sigmundsson, 2012).

In this sense, Moreno & Ordoño (2015) suggested that variability implies a load the organism needs to adapt to, and its effect on learning could be explained through the general adaptation syndrome (GAS) (Selye, 1956). In the adaptation of this syndrome to motor learning, it is hypothesized that when facing a practice load, an organism will experience a performance decrement (alarm phase). If the magnitude of the load is below a certain tolerance threshold, the organism will adapt to it (resistance phase). Nevertheless, intensities not adjusted to the individual characteristics of the learner would not cause the expected effect or would even damage performance. Consequently, one of the most relevant factors that needs to be considered when implementing variable practice interventions could be the learning or training process stage when the variability load is applied. According to Caballero et al. (2017), the effect of variable practice is not the same if the individual is at an initial stage of learning or if the participant shows a certain level at performing the task. Consequently, depending on these factors, the optimal variability magnitude will not be the same in all learning stages.

Therefore, it can be said that the effect of the application of variability load in the learning of a task is still unclear as it depends on many factors, such as the moment of application or the magnitude of the variability load. This may be related to the fact that the variability showed by the practitioner is directly connected with the effect elicited by an external variability load. From the DST point of view, the application of external variability after a constant training process could be a facilitator in the search for solutions, as it implies the modulation of external constraints. It would also fit in with the challenge point hypothesis prediction, as the nominal difficulty of the task should be adjusted to the performer’s level (functional difficulty). In this sense, one of the less studied factors in previous research is the moment in which the external variability load is applied (Lai et al., 2000). As beginners and experts seem to respond differently to variable practice, the analysis of this factor could provide a possible explanation to these differences. Thus, the present study has tried to analyze the effect of constant practice against variable practice, applying variability at different moments of the learning process according to the level of dexterity shown by the participants. The main hypothesis of this paper is that variable practice will be more effective after a constant training period, once the performance is stabilized.

Materials and Methods

Participants

Thirty-five healthy, right-handed adults initially took part in the study. During the experiment, three participants showed atypical results attributed to artefacts in the data collection, so thirty-two participants completed the experiment (n = 32), out of which nine were females and twenty-three were males (mean age 26.50 ± 5.93 years, body height 172.33 ± 7.58 cm, and body mass 68.87 ± 8.27 kg). None of the participants reported any kind of injuries at the moment of the research or in the previous 6 months. Written informed consent was obtained from each participant prior to testing. Data were treated anonymously, and all participants were informed of the risks and benefits of the trial. The experimental procedures used in this study were in accordance with the Declaration of Helsinki and were approved by the University Office for Research Ethics (DPS.FMH.01.16).

The participants were distributed into three groups according to the kind of training they were going to follow during the research: Constant, Variable and Stabilized (variability after a stabilization in the performance of the task) (See Table 1 for descriptive statistics of each group). The group distribution was established according to the initial performance of the participants in the test to avoid its influence on the final results.

Table 1 Descriptive statistics of the participants divided according to the kind of training.

Group	Age (years)	Height (cm)	Body mass (kg)	
Constant (n = 9)	27.25 ± 5.17	172 ± 4.92	67.63 ± 7.56	
Variable (n = 12)	25.82 ± 3.71	168.91 ± 9.47	67.91 ± 11.09	
Stabilized (n = 11)	26.63 ± 8.27	176 ± 5.66	70.73 ± 5.42	

Experimental procedure

The task consisted of a computer simulation in which the participant used a virtual ball and performed several throws trying to hit a mobile target that moved from one side of the screen to the other and back. To obtain a successful throw, the ball had to intercept the mobile target within a delimited area on the computer screen (the area limited by brackets in Fig. 1). The task was specifically designed by the researchers for this study using the Labview Software v.11 (National Instruments, Austin, TX, USA).

Figure 1 Task scheme.

The square represents the target, which moves from one side to another at a movement frequency of 0.25 Hz in each repetition. The participant had to throw the ball using the joystick, so that it impacts that target. The space delimited with brackets represents the effectivity area in which the ball should impact the target.

The throws were performed using a joystick connected to a dynamometer (model FSSB R3 Warthog, RealSimulator, Madrid, Spain), which registered the forces applied at the antero-posterior (AP) and medio-lateral (ML) axes at 100 Hz. The release angle of the ball was calculated by the resultant of the combination of the forces. The joystick was placed on a table at a 30 cm distance from the computer screen.

The participants had to perform the task with their non-dominant hand, seated on a chair in front of the screen and holding the joystick. They were told to direct their forearm towards the screen. The participants chose a comfortable sitting position in front of the screen, placing their other hand on their thigh (Fig. 2).

Figure 2 Position of the participant while performing the task.

Each participant performed a pre-test, a training period, a post-test and two retention tests (24 h and 1 week after the first day, respectively) (Fig. 3). All the tests consisted of three sets of 10 repetitions, with a 15 s rest between sets. Session 1 lasted 45 min, while sessions 2 and 3 lasted approximately 10 min. In all the tests, the movement frequency of the ball was constant (0.25 Hz). Before starting, all the participants were instructed about the functioning of the program, and they were allowed to perform five repetitions to get used to the program and the task. A training process was performed between the pre-test and the post-test. Participants performed different types of training according to the group they were assigned to. (a) Constant training (Constant): participants performed the same task as in the evaluation tests, in the same conditions. (b) Variability training (Variable): in this group, a non-structured variability practice at the objective level of the task was applied (Ranganathan & Newell, 2013). The modified parameter between repetitions was the movement frequency of the target, which varied in a random pattern between 0.125 and 0.5 Hz. According to Davids, Button & Bennett (2008) and Ranganathan & Newell (2013), variability practice at the objective level encourages the ability to look for the best options to achieve the task goal. (c) Variability after the performance stabilization during the learning of the task (Stabilized): in this group, the software was set up so the participant started with constant training, and the performance improvement was registered. After the seventh set of 10 repetitions, the error displayed by the participant in the last two sets was compared with the error from the previous sets (by calculating the difference). If any of the two last sets displayed a lower error than the previous, the participant continued the constant training. Once the participant did not reduce the error in two consecutive sets, the variability training started (in the same way as the Variable group) and continued until the end of the training process. This was an individualized process, in which each participant started variability training at the improvement plateau. Researchers decided not to apply any variability load until the seventh set was reached to allow, after the pre-test, at least a minimum series of three additional sets to assess the stabilization in the evolution of performance. The number of sets performed was the same for all the groups (44 training sets of 10 repetitions with a 15 s rest between sets). The temporal distribution of the tests is shown in Fig. 3.

Figure 3 Temporal distribution of the tests.

Data analysis and reduction

The final position of the ball was registered by the software used to record the task performance (developed in LabVieW v.11). The absolute value of the distance between the ball and the target (AE) was taken as the main performance indicator. The variable error (VE) was measured as the standard deviation of the final position of the ball. The software computed the absolute values of the distance between the ball and the center of the delimited effectivity area in which the ball should impact the target (ball-to-area), and it also computed the absolute values of the distance between the target and the center of the effectivity area (Target-to-area) as complementary information. An application written in MatLab R2020a (MathWork, Inc., Natick, MA, USA) was used for data analysis.

Participants were distributed into three groups according to the kind of training they followed (Constant, Variable and Stabilized). Additionally, they were subdivided according to the initial VE of the participants during the pre-test, low or high. The value of VE reflects the outcome variability that the participants exhibited during the execution of the required task. As the variability in the outcome is an intrinsic characteristic of the individual, this variable was selected to analyse how it could be related to the effect of variable practice. Those participants who displayed lower values of this variable than the mean of the sample were classified as low outcome variability (LV, n = 16). Participants classified as high outcome variability (HV, n = 16) were those who showed over-the-mean values of VE (Constant: n = 9, LV n = 4, HV n = 5; Variable, n = 12, LV n = 5, HV n = 7; Stabilized, n = 11; LV n = 7, HV n = 4). The analysis was to explore how the effect of different kinds of training would be related to the intrinsic features of each participant (Caballero et al., 2017; Chrousos, 2009; Garcia-Herrero et al., 2016).

Statistical analysis

Prior to the statistical analysis, a Kolmogorov–Smirnov test confirmed the normal distribution of the data. After that, a two-way (4 × 3) mixed ANOVA was carried out. The intragroup factor was the moment of measurement (four levels: pre-test, post-test, re-test 1 and re-test 2), and the intergroup factor was the type of training practice (Constant, Variable, and Stabilized). The interaction effect of the type of training practice on the differences between the moment of the measurement was also assessed. A Bonferroni post hoc for the pair comparisons was carried out for the different group divisions, the kind of training group and the initial outcome variability group.

The subdivision according to the initial outcome variability was not included as a factor due to the reduced sample size, but the effect sizes (g) were calculated applying the Hedge’s correction to minimize the impact of the sample size and they were categorized following Rhea’s standards (Rhea, 2004) for non-trained individuals. Additionally, a correlational analysis was also carried out to test the relationship between the initial outcome variability and the improvement percentage between different testing moments. The signification level and the Pearson correlation coefficient were calculated.

The significance level for all the analysis was set at p < 0.05.

Results

The results of the statistical analysis are presented in the following lines. The ANOVA showed significant main effects for the moment of measurement factor. All the groups showed an improvement after the practice period (AE: F3,87 = 26.43, p < 0.001, η2 = 0.477; VE: F3,87 = 32.30, p < 0.001, η2 = 0.53; Target-to-area: F3,87 = 757, p < 0.001, η2 = 0.21; ball-to-area: F3,87 = 8.70, p < 0.001, η2 = 0.23), but without significant intergroup differences. No interaction effect was observed (AE: F6,87 = 0.17, p = 0.98, η2 = 0.01; VE: F6,87 = 0.60, p = 0.68, η2 = 0.04; Target-to-area: F6,87 = 0.29, p = 0.81, η2 = 0.02; Ball-to-area: F6,87 = 0.28, p = 0.82, η2 = 0.02).

The comparisons between tests in all the variables for the three intervention groups (Constant, Variable and Stabilized) show statistically significant results in AE and VE (Fig. 4). Taking AE as the key performance indicator, the three training groups showed significantly better performance in the post-test and the re-test2 compared to the pre-test, with a moderate effect size (Constant group: g = 1.56 and g = 1.27, respectively; Variable group: g = 1.43 and g = 1.33, respectively; Stabilized group: g = 1.63 and g = 1.12, respectively).

Figure 4 (A–D) Pairwise comparisons between tests in all the variables for the three intervention groups (Constant, Variable and Stabilized).

a, significant differences compared to the pre-test, p < 0.05; A, significant differences compared to the pre-test, p < 0.001.

The evolution of the error and its dispersion among the three training groups is presented in Fig. 5. In the constant group, a progressive decrease can be seen (with typical transient increases and decreases), especially at the beginning of the mean error values during the training series. The error values of the post-test are lower than those of the pre-test. Despite the slightly higher values in the first retention test, these did not reach the pre-test values. In the second retention test, the error values were similar to those of the post-test. The behaviour of the standard deviation is very similar to the error values in this group.

Figure 5 Evolution of the absolute error (AE) and dispersion (SD) of the participants of the three training groups: Constant, Variable and Stabilized. Note: Mean = AE.

The variable group did not show a clear trend in performance during the training series. An unstable pattern of error and, more clearly, of its dispersion can be observed. It can be seen how the error remained constant, with some peaks and valleys, during all the training process, the same as the dispersion, which showed a marked variable behaviour during the whole practice period. Nevertheless, the post-test results, with lower error values than in the training series and the pre-test, would elucidate latent improvement in performance. This is corroborated by the values of the mean error in the retention tests. The variability in the error, expressed by the dispersion values, followed a similar behaviour with lower differences.

The stabilized group, showed a marked decrease both in mean error vales and in its dispersion at the beginning of the training. Then (coinciding with the application of variable practice), the error increases and, at one point, it starts decreasing again progressively. This increase in the mean error values is not seen in the SD, which maintains a more constant trend, although it fluctuates slightly more at the end of the practice period. The behaviour in the post-test and retention tests was similar to the variable group.

A correlational analysis was carried out to test the possible relationship between the initial variability in the outcome and the improvement in performance (error reduction) due to the training. This improvement was computed by calculating the difference between the initial error in the pre-test and the three tests measured after the training (post-test, re-test 1, and re-test 2). Higher values in improvement mean greater decrements in AE. Statistically significant correlations were found between the initial VE and the improvement in AE after the training phase (difference between the pre-test and the post-test), r(30) = 0.474, p = 0.006. Initial VE also correlated with the improvement shown in the re-test 2, r(30) = 0.468; p = 0.007. The correlation between the initial outcome variability and the improvement shown in the re-test 1 was not significant.

Regarding the comparisons made within the subjects of each group according to their initial VE, we can see some remarkable size effects (g) according to Rhea’s criterion (Rhea, 2004). In the constant training group, the LV participants showed a moderate effect size (g = 1.21) between the pre and the post-test. That difference was higher in the HV participants, showing a large effect size (g = 2.29). The effect size of the differences between the pre and the re-test 1 and 2 were low for the LV participants (g = 0.92, g = 0.94) and moderate for the HV participants (g = 1.87, g = 1.80). In the variable training group, it can be seen that, conversely, the effect size was larger in the LV participants than in the HV participants in the pre-test vs. the post-test (LV: g = 2.75, HV: g = 1.17) and in the pre-test vs. the re-test1 (LV: g = 2.04, HV: g = 0.96). These effects changed in the comparison the pre-test vs. the re-test2 (LV: g = 2.86, HV: g = 4.27). In the stabilized training group, there were less differences between the LV and the HV participants in the effect of the training (LV: pre-test vs. post-test: g = 4.70; pre-test vs. re-test1: g = 2.21; pre-test vs. re-test2: g = 2.79; HV: g = 2.29; g = 0.96; g = 4.27).

Discussion

The main goal of this study was to assess if variable practice is beneficial for learning and if the controversial results observed in the previous literature can be explained by the incidence of other variables, such as the moment when variable practice is applied or the individual characteristics of the learner.

As it was expected, within the measured variables, the variable that was more clearly affected by training was the absolute error, which was the main performance variable. All groups improved their performance after practice, without there being clear differences between the different training strategies. Moreover, we can observe that the different training strategies led to different responses in the improvement of the participants. Although all the participants improved, the larger effect sizes shown by the participants in the variability training group suggest that the addition of variability in practice implies a beneficial training load (Moreno & Ordoño, 2015; Savelsbergh et al., 2010). However, the lack of clear differences between groups and the absence of interaction effects hinders the obtention of clear conclusions about the benefits of variable practice and it reinforces the need to discuss what factors could mediate in the relationship between variable practice and performance.

The introduction of variable practice once the participants stabilized their performance was manipulated as one possible factor that could affect the differential effects of variable training. This is based on the idea that the amount of variability applied in a session needs to be matched to the skill level of the performer, in line with the challenge point framework (Guadagnoli & Lee, 2004). Dhawale et al. (2019) suggested that, at the beginner level, low-task variability could be initially beneficial to guide exploration towards finding a functional solution. As the level of the learning increases, higher variability would be present in the individual, in the task and in the environmental constraints to promote a more dexterous behaviour. Previous studies pointed out that individualization is a fundamental issue, and the characteristics of the individuals should be considered when variability is applied (Caballero et al., 2017). In this experiment, it was expected that a previous period of stabilization by practicing under constant practice would prepare the learners to exploit the benefits of variable practice better. Once the initial functional solution was acquired, variability would encourage exploration of new successful and adaptive solutions to the task. Nevertheless, the absence of between-groups differences has encouraged the need for a more in-depth exploration of how the differences between learners inside the groups would mediate the effect of variable practice. Barbado et al. (2017) proposed that the learners with higher initial variability would exhibit a greater exploratory behaviour that allows them to learn in a greater way. This higher initial variability would reduce the need for the enhanced variability proposed by the practitioner. This can be related to the challenge point hypothesis that proposes that, when the difficulty of a task increases, the available knowledge about the task has to increase in parallel to enhance the learning process, especially when the level of the practitioner is low, being less necessary when the level increases (Guadagnoli & Lee, 2004).

It can be seen how constant practice did not result in great variations of performance during the training process, opposed to variable practice, which caused an irregular behaviour in the performance during all the performance series. Moreover, in the stabilized group, the trend towards the increase of the error after stabilization would indicate the effect of external variability application. In this group, a performance increase can be seen at the end of the training period, indicating an adaptation after the alarm state caused by the variability application (Moreno & Ordoño, 2015; Selye, 1956). Moreover, the performance values in all the tests were similar in all kinds of training, which can indicate that, in general terms, all the different ways of training enhanced performance.

Focusing on the correlational analysis, the individuals who showed a greater outcome variability (i.e. greater VE) in the pre-test displayed greater improvements than those who were initially more consistent. This is consistent with the fact that those participants with poorer initial performance had more room for improvement. Thus, it could be possible to interpret that the individual’s performance level should be taken into consideration to decide what kind of training strategy will be more suitable for his progression (Caballero et al., 2017; Garcia-Herrero et al., 2016). Supporting this idea, the effect size results could indicate that participants with higher outcome variability could benefit more from less external variability, and the opposite would happen with more consistent participants, while the moment of application could play a key role here (Lai et al., 2000). This lays the basis for future investigations with a higher number of participants which can be divided according to their initial performance and variability.

In the constant training group, the participants with higher initial error variability increased their performance with clearer changes. These participants also showed poorer initial performance and, therefore, more room for improvement was expected. However, it has to be noted that their error values at the post-test and re-tests were similar to those obtained by the learners with lower variability in their initial outcome, suggesting that these last participants did not exploit the benefits of this kind of practice. In the group that trained with variable practice from the beginning, the participants with lower initial error variability reduced their error to a greater extent in the post-test and in the first retention test than the participants with high initial error variability. In this sense, Caballero et al. (2017) proposed that the training load should be adapted to the individual’s level. Barbado et al. (2017) and Garcia-Herrero et al. (2016), also postulated that the learners that showed a more variable performance could have an exploratory behaviour which allows them to learn in a greater way. However, in both cases (high and low initial error variability participants) there was a progressive improvement during the training process. More information is needed about the effect of additional variable practice when learners exhibit low levels of variability.

All the participants exhibited a decrement in their performance during the application of variable practice. However, the removal of variable practice resulted in a considerable improvement of the performance. This could be explained by the adaptation to the training load, as proposed by Moreno & Ordoño (2015). The applied variability caused a transitional loss of performance in participants, who improved when they practiced without enhanced variability and after a resting period. Previous studies suggested that poising the system above a “critical point” of the edge of chaos (Bowes & Jones, 2006), in which the amount of variability would lead to excessive instabilities, would limit the search for new solutions (Dhawale et al., 2019).

Finally, the fact of applying variability after stabilizing the performance of the participants with constant practice did not show a differential improvement in the final result compared to variability training from the start of the process. Applying constant training to facilitate stabilization was not reflected in an improvement of performance. This could be especially useful for those participants with high initial error variability. Again, the application of variable practice resulted in a reduction in performance. When variable practice was removed, in the post-test, the performance increased, and this improved performance continued in the retention tests, but with similar levels to those obtained during the stabilization series.

The implications of the results in this experiment should be taken with caution due to the sample limitations of the study. It is important to consider that this task was completely new for the participants, and the sample showed heterogeneous results. Although the results indicated different training impacts in each participant depending on their inner features, the reduced sample size in each subgroup hinders statistical between-group differences. The enlargement of the sample in further research is encouraged to be able to perform a more in-depth analysis and to obtain clearer conclusions.

In addition, other interesting variables could be registered to try to obtain additional information about the participant’s exploration, such as the decision time since the stimulus appeared until the participant responded, as this factor could have some influence on the precision of the task. In this study, only the variability in the outcomes has been studied. Recent studies have outlined that the role of task-relevant variability could be implied in faster learning (Haar, van Assel & Faisal, 2020). Thus, this lays a new field of research for future studies. Additionally, it must be pointed out that the tests were all applied in constant conditions, and a transference test was not included. Previous studies have supported the good results of variable practice in transference tests compared to constant training (Pacheco & Newell, 2018). Although it was not the aim of the study, this type of tests could have provided additional findings according to the strategy of practice, so it could be taken into consideration in future research.

Conclusions

After the analysis of the results, it can be concluded that the practice of the task has improved performance in all cases. Variable practice did not cause a significantly greater improvement when compared to constant practice. Applying variable practice after a stabilization period training with constant practice did not induce a significant improvement either. Regarding the decrease in performance after the variable practice load as an adaptation phenomenon (Moreno & Ordoño, 2015), this decrease could only be appreciated when the stabilization period, using constant practice, was applied. Nevertheless, the effect of variable practice seems to be modulated by the initial outcome variability of the participants. Those participants with higher initial variability values showed higher improvement trends than those with lower values when constant practice was applied. However, participants with a smaller variability in their outcome did not show this positive effect of the constant practice in their performance.

The stabilization of the performance period after constant practice applied to high initial variability learners did not show a positive effect that prepared them to face a variable practice load. It is still unknown if longer constant practice periods would be effective to prepare the individuals with high variability to benefit from variable practice effects. Further studies are needed to study more in-depth on how to adjust the practice conditions to the participant’s characteristics.

Supplemental Information

Supplemental Information 1 Scattering plots between the initial variability (VE in the pretest) and the improvement percentage between the pretest and the different testing moments after training.

Click here for additional data file.

Supplemental Information 2 Main effects of moment of testing in the variables measured in the two-way repeated measures ANOVA.

Click here for additional data file.

Supplemental Information 3 Results of all the test performed by the participants.

- Group (1=Constant training, 2= Variable training, 3=Variability applied after performance stabilization)

- Initial Variability: 1=low; 2=high The database includes all the means (M) and standard deviations (SD) of all the measured variables in the Pretest, postest, retest1 and retest2. The variables ending in 3 correspond to the mean data of the 3 performed series of the task. The variables ending in 2_3 correspond to the mean of the two best results out of the 3 performed series.

Click here for additional data file.

Supplemental Information 4 Statistical results.

Click here for additional data file.

Supplemental Information 5 Codification of the variables shown in the dataset.

Click here for additional data file.

Additional Information and Declarations

Competing Interests

Author Contributions

Human Ethics

Data Availability

The authors declare that they have no competing interests.

Carmen Ruano conceived and designed the experiments, performed the experiments, analyzed the data, prepared figures and/or tables, and approved the final draft.

Carla Caballero conceived and designed the experiments, analyzed the data, prepared figures and/or tables, authored or reviewed drafts of the article, and approved the final draft.

Francisco J. Moreno conceived and designed the experiments, authored or reviewed drafts of the article, and approved the final draft.

The following information was supplied relating to ethical approvals (i.e., approving body and any reference numbers):

The experimental procedures used in this study were in accordance with the Declaration of Helsinki and were approved by a University Office for Research Ethics (DPS.FMH.01.16).

The following information was supplied regarding data availability:

The results obtained in the study are available in the Supplemental Files.

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
