# Peer review of "The effect of enhanced variability after performance stabilization through constant practice"

_PeerJ, doi:10.7717/peerj.13733_

## Round 0.1 · original submission · Major Revisions

After careful consideration, we feel that the manuscript has merit but does not fully meet the publication criteria as it currently stands. Therefore, we believe the manuscript is under the interest and would benefit from a major revision. Authors should improve the points raised during the review process.

Reviewer 1 ·

Basic reporting

The authors should make a clear distinction between “variable practice” (more in line with Schmidt’s schema theory, where the experimenter introduces variations of the task, usually by changing the task goal) from the “intrinsic” motor variability (which is tied to exploration in Wu et al. etc.). Not making this distinction clearly in the Introduction can be confusing for the reader.

Statistics value in the Results section should also mention the degrees of freedom of the test. – i.e. (F(x,y) = 27.47, p < 0.001)

Experimental design

The task is not fully explained clearly – rather than a time, the movement of the target is described in terms of a frequency. Did that mean that participants could watch the target move back and forth and release the ball at any time they wished? Or did they have to do it on the very first time it reached the target zone? IF it is the former, it is important to also note the ‘decision time’ – i.e., the time that the participants waited before releasing the ball because that could have an influence on the accuracy

Validity of the findings

The “subgroup” analysis is too small to be of value (with 10/group, each group has only 5 participants). If the authors wish to pursue this analysis, they should use a correlation metric instead of arbitrarily dividing a continuous variable into two groups.

The authors should also provide data during the training phase – this would inform the reader what the effect of introducing variability on practice was. Without this information, it is not possible to interpret the retention tests (for e.g., if the variable practice did not create significant increases in errors, then the task could have been ‘too easy’ and therefore could explain why there is no difference between the groups)

The authors use different dependent variables all related to the performance. This is risky since it increases the chance of false positive (as there are multiple comparisons being made). The authors should pick the one that is closest to the instruction to the participant as a primary variable – for example, if the participants were told to hit the target as accurately as possible, then they should report the AE as the primary variable that is used to make the conclusion. The other variables should be mentioned only if they provide additional insight.

Additional comments

I am writing my entire review to provide authors context of what I view to be my major concerns with the current manuscript:

The goal of this study was to examine the effect of variable practice and whether introducing variable practice after stabilizing performance has a beneficial effect on learning. The overall results showed that there was no significant difference between the groups, although there was some difference based on the inherent variability in the learner (i.e., whether they had initially higher or lower variability to start with)
Although this is an important issue, there are several missing pieces of information that do not allow to evaluate the results fully. I have explained my concerns below:
Major:
The “subgroup” analysis is too small to be of value (with 10/group, each group has only 5 participants). If the authors wish to pursue this analysis, they should use a correlation metric instead of arbitrarily dividing a continuous variable into two groups.

The authors should also provide data during the training phase – this would inform the reader what the effect of introducing variability on practice was. Without this information, it is not possible to interpret the retention tests (for e.g., if the variable practice did not create significant increases in errors, then the task could have been ‘too easy’ and therefore could explain why there is no difference between the groups)

The task is not fully explained clearly – rather than a time, the movement of the target is described in terms of a frequency. Did that mean that participants could watch the target move back and forth and release the ball at any time they wished? Or did they have to do it on the very first time it reached the target zone? IF it is the former, it is important to also note the ‘decision time’ – i.e., the time that the participants waited before releasing the ball because that could have an influence on the accuracy

The authors use different dependent variables all related to the performance. This is risky since it increases the chance of false positive (as there are multiple comparisons being made). The authors should pick the one that is closest to the instruction to the participant as a primary variable – for example, if the participants were told to hit the target as accurately as possible, then they should report the AE as the primary variable that is used to make the conclusion. The other variables should be mentioned only if they provide additional insight.


Other:
The authors should make a clear distinction between “variable practice” (more in line with Schmidt’s schema theory, where the experimenter introduces variations of the task, usually by changing the task goal) from the “intrinsic” motor variability (which is tied to exploration in Wu et al. etc.). Not making this distinction clearly in the Introduction can be confusing for the reader.

Statistics value in the Results section should also mention the degrees of freedom of the test. – i.e. (F(x,y) = 27.47, p < 0.001)

Ln 225 – “Significative” should be “significant”?

Since the stabilized group changed to the variable condition based on an individual criterion, it would be useful to report at what practice session individual participants switched to the ‘variable’ task and whether this influenced the error

What is the exact number of participants in the high and Low initial variability (LV) within each group?

Reviewer 2 ·

Basic reporting

• Clear and unambiguous, professional English used throughout.
I am not an English native speaker. Globally the article is easy to read but the structure of some sentences suggests that the authors are also not native speakers.
• Literature references, sufficient field background/context provided.
The literature review should be extended, notably regarding the role of variability in learning.
The first paragraph of the introduction seemed to place the article within the Dynamic Systems Theory, but the literature review is lacking results from this field that could improve the rationale leading to the method of the article. For example, the conclusion of the introduction (l.89-91) mentions that the moment of application and the magnitude of the variability are factors influencing the effect of variability on learning. However, the first factor is not well developed in the literature review, as only one study is presented (l.84-86). Lai, Shea, Wulf, & Wright 2000 (experiment 2) may be relevant as it presents a study design similar to the present article, but the aim of this study was to test some predictions from the Schema theory. I suggest that the introduction provides a rationale that explains why variability after stabilization would be suitable from a DST perspective. Regarding the second factor (i.e., load magnitude), the literature review does not appear necessary as the aims of the study are not related.
The rationale of the study is partly justified in one paragraph of the discussion (l.280-302). This paragraph would be more suitable in the introduction as it presents the Challenge Point Framework (which is not mentioned in the introduction) and explains why late variable practice would be effective. Moreover, this paragraph does not discuss the results of the study.
Also, a part of the literature review (l.60 to 66) refers to articles testing the contextual interference hypothesis and the schema theory to address the effect of variability in practice on learning in different populations (children and adults). However, these two approaches present clearly distinct learning protocols: contextual interference protocols are designed to learn multiple tasks simultaneously (usually with a random vs a blocked schedule), whereas schema theory proposes to vary some parameters within a single task (usually involving constant vs. variable practice). In the context of this article, the literature about contextual interference does not fit the study aim.
There are some issues with the references. I saw that some years differ between the text and the reference section (e.g., Williams & Hodges, 2015 in text and 2005 in references, line 40) and some years are missing in the text (e.g., Caballero et al., line 51).
• Professional article structure, figures, tables. Raw data shared.
The structure of the article is clear.
The definition of the figure 4 could be improved. Figure 4, 5 cannot be understood without the text as some presented variables are not defined in the legend. For example, the title of the second panel in figure 4 (“AE VE Values (cm)”) cannot be understood as the reader doesn’t know to what “AE VE” stands for.
The shared data (peerj-64066-Database_means&SDs_-_modificada.xlsx) are not “raw” as they are the mean and standard deviations of the dependent variables. I do not understand to which variable the columns of the file correspond to and some values appear to be missing. Also, some of the data are presented in Spanish in the Resultados.zip file.
• Self-contained with relevant results to hypotheses.
The article represents an appropriate unit of publication.

Experimental design

• Original primary research within Aims and Scope of the journal.
This article is original research.
• Research question well defined, relevant & meaningful. It is stated how research fills an identified knowledge gap.
The research questions are very relevant and need to be addressed.
• Rigorous investigation performed to a high technical & ethical standard.
The ethical standards seem to be respected. The data collection and analysis appear to be rigorously performed. The statistical analyses should be further explained, and the results should be reported in more details.
• Methods described with sufficient detail & information to replicate.
The “material and methods” section is well detailed.

Validity of the findings

• Impact and novelty not assessed. Meaningful replication encouraged where rationale & benefit to literature is clearly stated.
The research question is very relevant and needs to be addressed. Notably, the investigation of the influence of learners’ intrinsic variability on the effect of variable practice conditions could help to better understand the individual differences observed in the motor learning literature. However, the measure of intrinsic variability is not adapted.
• All underlying data have been provided; they are robust, statistically sound, & controlled.
As mentioned above, raw data are not provided, and the current state of the data files is not clear.
The statistical analyses should be detailed.
• Conclusions are well stated, linked to original research question & limited to supporting results.
The conclusions, and more globally the discussion, are not clear. For example, two opposite conclusions are made in the second paragraph (l.269-279: there are no group differences, but variable practice had a positive effect on participants ‘performance). Moreover, theoretical aspects are discussed (like the Challenge Point Framework) but they were not presented in the introduction.

Additional comments

The first aim of this study is to investigate whether applying variable practice conditions after performance stabilization could improve motor learning. The second aim is to investigate whether the intrinsic variability of the participants modulates the practice effects. The study design involves three practice groups: a constant practice group, a variable practice group and a constant-variable practice group. The task was a virtual aiming task performed on a computer with a joystick. Performances were measures during pretest, training, post-test and two retention tests. No significant between-group results were found.
The research questions are very relevant in the field of motor learning. The methods section is well presented and clear and the data collection and analysis appear to have been performed rigorously. However, I have three major concerns regarding this article: i) the literature review needs to be deepened and more related to the current study to better support the methods used, ii) the measure of intrinsic variability is not adapted and iii) the statistical analyses are unusual and do not appear suitable to respond to the study aims. In what follows, I will present in detail these concerns and then some detailed comments.
The literature review does not support the rationale of the study. As stated in the “basic reporting” of the section, some of the literature presented is not relevant (e.g., references about Contextual interference). Also, some aspects of the study design are not justified in the introduction but sometimes elsewhere: i) the nature of the variability is justified in the methods l.157-161 but it is not presented as one of the factors influencing the effect of variable practice in the introduction and ii) the rationale supporting the design of the stabilization-variable practice is explained in the discussion l.280-302). Also, the literature review about the effect of the magnitude of the variability is not necessary in the article as it was not manipulated in the study. Similarly, I did not understand why the paragraph about training load is relevant regarding this study (l.67-80).
The measure of intrinsic variability does not seem adapted. It is measured with the variable error of the participants (l.183-184), that is the standard deviation of the distance between the ball and the target. This measure reflects the variability in outcomes but not the variability of the movement system. For example, Haar, van Assel & Faisal 2020 also explored the link between movement variability and learning in an aiming task (pool billiard) and their results suggested that only initial variability in task-relevant joints could predict learning. Thus, in line with my previous concern, the article should define “intrinsic variability” in the introduction and present some previous studies which investigated the relationship between “intrinsic variability” and learning. Also, the number of learners included to the LV and HV groups is not indicated.
I do not understand some choices in the statistical analyses and the results are not fully reported. From my reading, the mixed-ANOVAs performed were not mixed because the two factors are two between groups factors (the practice condition: constant, variable and constant-variable; and the initial variability: LV and HV). But in the results (l.201-207) the factors appear to be one between group factor (the practice conditions) and the tests (but the tests included in the analyses are not presented). Moreover, why are the three factors (test – practice condition – initial variability) not integrated to a three-way ANOVA directly? This needs to be clarified. Then, the report of the results of the ANOVAs is incomplete as the F and p values are not always reported. It is also surprising to see that pairwise comparisons between practice conditions were performed although the ANOVAs did not show any significant effect of the group factor. Similarly, the subsequent computation of the effect sizes appears unusual as they are presented like pairwise comparisons, and they are performed on dependent variables that were already used in the ANOVAs. However, I am not familiar with this kind of analysis (the computation of the g effect size and its interpretation).
Detailed comments:
- Why were 31 participants recruited? The sample size may explain the absence of significant difference between groups. Why were the group unbalanced (i.e., 9-11-11)? Did some participants drop out?
- How long did the session on day 1 last? This information would be relevant for replication.
- In the figures, “posttest” and “re-test” are written differently (with one or two “t”). In the text, “retests” are also referred to as “retention tests”. This could be harmonized throughout the article.
- The information that the target was moving from side to side at 0.25Hz is redundant in the article. Moreover, this movement frequency is the one used for the constant practice condition, but it is varied in the other conditions. Thus, the frequency should not be indicated at the beginning of the experimental procedure (l.128).
- The task was designed specifically for this study (l.130). A recent review pointed out that too many tasks are used in the motor learning literature which prevent replication of study and comparisons of results (Ranganathan, Tomlinson, Lokesh et al., 2020). Therefore, knowing that virtual aiming task with a joystick (e.g., Ranganthan and Newell 2010 in Experimental Brain Research and in Journal of Motor Behavior the same year) were already developed, why was this new task designed?
- The description of the participants’ arm position (“forearm and arm aligned” l.137-138) does not correspond to the subject’s arm position in the figure 2.
- There is no verb in the sentence l.155.
- Why does the sentence l.159-161 initially refer to Ranganathan and Newell (2013) and to Davids et al. (2008) at the end?
- Why could not the variable practice start before set 7 (l.164)?
- How was it assessed that the participant “was not able to improve” l. 164-165?
- Usually, learning is also assessed with a transfer test, notably when variable practice is involved. Why were only retention tests performed? Why not testing the participants with a new frequency to assess if variable practice facilitated the generalization of learning? We may expect better transfer of learning for the group in variable practice according to Schema theory and recent results (e.g., Pacheco & Newell, 2018) whereas I don’t see a theory that would state that retention would be eased with variable practice.
- A “)” is missing l.206.
- To which test do the results on VE l.218-219 refer to?

---

## Round 0.2 · Major Revisions

The manuscript has merit but does not fully meet the publication criteria as it currently stands. The first reviewer states that authors have addressed most manuscript issues; however, reviewer two is still rising theoretical and methodological concerns.

We believe the manuscript is under interest and would benefit from a major revision. Authors should improve the points raised during the review process.

Reviewer 1 ·

Basic reporting

The authors have addressed most of the concerns from my previous review. I still have a few concerns:

1. The VE of the absolute error is not equivalent to the variability (since the AE removes all information about the sign). This measure should either be computed as a true VE (i.e. where the sign of the error is retained during computation) or removed
2. In Figure 5, the authors should plot the groups on the same plot so that comparison is easier. Also the line with the SD should be plotted as an error bars (and not a separate line)
3. Ln 283 – the authors should present the scatterplots associated with these correlations
4. It looks like the authors had indicated that they were using some service for proofing. I was not sure if the current manuscript has already gone through this proofing but it would be helpful to correct some minor grammatical errors throughout the manuscript

Experimental design

no comment

Validity of the findings

no comment

Additional comments

no comment

Reviewer 2 ·

Basic reporting

1. Clear and unambiguous, professional English used throughout.
/
2. Intro & background to show context. Literature well referenced & relevant
The literature review in the introduction is still lacking consistency to explain the rationale leading to the developed method.
First, the theoretical background used is unclear. Although the DST is mentioned more often in this new version of the manuscript, none of the concept or tools from this theory are used. Moreover, some studies investigating movement variability are cited in the paragraphs about DST, although they do not refer to this theoretical approach (for example Wu et al. 2014 refer to reinforcement learning theory, Herzfeld & Shadmehr, 2014 are also referring to reinforcement learning – moreover, they do not present an original study as suggested l.48). Similarly, Challenge point hypothesis is cited in relation to DST although the challenge point hypothesis has no links with DST.
Second, the definition of variable practice remains unclear. Schema theory is cited (l.41) to present this perspective of variable practice and the rationale explaining the benefits. But, in what follows (l.43-46), the DST is mentioned without concretely explaining why from a DST perspective inducing variability during would be beneficial to learning. Maybe presenting the rationale from the cited studies could help to explain this point. Moreover, the meta-analysis by Brady (2004) about contextual interference is still used in the introduction (l.78-81), although I already pointed out in the previous revision of the manuscript that contextual interference is different from variable practice.
Third, the paragraphs about “practice load” (l.55-59 and 84-97) are still unjustified regarding the study aims. Also, I did not perceive the link with DST.
Finally, and most importantly, the definition of “intrinsic variability” added in the introduction (l.32-37) does not fit how it was measured experimentally. In the introduction, “intrinsic variability” refers to “movement variations” but it was evaluated with the VE score in pretest of the aiming task, that is the variability in the distance between the ball and the target, which does not reflect the movement performed by the participants. regarding the task and the given definition, “intrinsic variability” would rather refer to variations in the movements performed with the joystick (e.g., the force applied or the direction of the force application) regardless of the movement outcomes (i.e., the distance between the ball and the target). According to the response of the authors to my previous comments, variability in the force applied was measured and analyzed but it did not show significant results, so it was not reported in the manuscript. I suggest that it should be reported as the measure of “intrinsic variability” if the current definition is used.

In summary, based on the current form of the introduction, the rationale leading to the design of the experiment and the data analysis remains unjustified.

Additional comments:
l.49: Cardis et al., 2018 showed the opposite to what is stated, that is whatever the amount of variability, learning was impaired

3. Structure conforms to PeerJ standards, discipline norm, or improved for clarity.
/
4. Figures are relevant, high quality, well labelled & described.
The quality of the figure can be improved but the current quality may be sufficient to.
Figure 4 seems to be distorted and the last panel is not aligned with the others. I also suggest naming the 4 panels (e.g., A, B, C and D). Also, there are no significant differences reported in the target-to-area panel and the ball to area panel whereas some significant differences are reported in the results section.
Figure 5 can be improved by presenting the standard deviations with error bars around the means. Moreover, the reader does not know to what the dot are referring to (a session, a trial, or a block). Finally, the dispersion (SD) seems to be equivalent to what is presented as VE in the text, but I am not certain. This should be clarified.
5. Raw data supplied (see PeerJ policy).
Raw data are now available in the Database_(i)_(1).xlsx file.

Experimental design

1. Original primary research within Scope of the journal.
/
2. Research question well defined, relevant & meaningful. It is stated how the research fills an identified knowledge gap.
As highlighted in my main comment, the research question is interesting but the rationale leading to the experimental design and the data analysis remains not sufficient and seems partly not adapted to support the method used.
3. Rigorous investigation performed to a high technical & ethical standard.
/
4. Methods described with sufficient detail & information to replicate.
The sample size is still not justified, although it may explain that most of the results are not significant.
I still don’t understand how the stabilization of the performance was evaluated. The method section only state that variability training started “once the participant did not improve in two consecutive sets”. This should be developed.

Validity of the findings

1. Impact and novelty not assessed. Meaningful replication encouraged where rationale & benefit to literature is clearly stated.
/
2. All underlying data have been provided; they are robust, statistically sound, & controlled.
The addition of the descriptive results reported l.262 to 282 is welcomed. However, a more robust statistical analysis would be appreciated to properly compare the training dynamics of the different groups. For example, a 2 (early and late training phase) x 3 (groups) mixed ANOVA would be more informative.
The results of the correlations should report the degrees of freedom.
The effect size associated to the results of the ANOVAs could be reported.
Finally, I reiterate that I am not familiar with the computations of the effect sizes (g) presented like pairwise comparisons. Thus, I cannot judge the validity of the findings associated with these parts of the results.

Additional comments:
l.284: how are “the improvement in performance due to the training phase” measured? I have the same question about “the improvement between tests”. I could not find this information Did all the participants “improve” in both cases?


3. Conclusions are well stated, linked to original research question & limited to supporting results.
The conclusions presented in the discussion and the conclusion sections remain unclear.
First, there are no discussion of the literature associated with DST, which once more suggest that mentioning it in introduction is irrelevant.
Second, opposite conclusions are still made. For example, l.431 “Variable practice has not had a significantly higher improvement when compared to constant practice” whereas it is stated l.314 that “Those training groups which followed both variable practice schedules showed a larger improvement, regarding the effect size, than the individuals who followed a constant training”.
I suggest that the discussion should be shortened to focus on responding to the research question.
Additional comments:
l.313: The formulation of the sentence is confusing. What could have improved the effects of variable training?

Additional comments

In summary, I consider that this new version of the manuscript was not improved in comparison to the previous one.
As in my precedent review, I believe that the rational do not justify the experimental design of the study. Also, the measure of intrinsic variability used does not fit the given definition. Therefore, the current conclusions are invalid.

---

## Round 0.3 · Minor Revisions

One reviewer has some further minor suggestions for revision. Please address these in the manuscript and a response letter.

Reviewer 2 ·

Basic reporting

Regarding my previous concerns, the rationale of the study was improved in the introduction and the manuscript provides a clearer presentation of variable practice, which better justify the literature about practice load.
However, my main concern about “intrinsic variability” (now referred to as “initial variability”) remains still. Indeed, the use of “initial variability” throughout the text is confusing. In the sentence l.402-403 it refers to “Variable Error” which is the variability in performance outcomes. However, references to “variability” in the introduction refer to movement variability (for example l.67 with the reference to Caramiaux et al. 2018 who measured movement variability, or l.102 with “motor variability”). “Variability” to “exploration” are also linked in the authors’ response to my previous comments, and in the discussion (l.429-431), which, again, refers to movement variability.
Thus, the measure of “Variable Error” in pre-test still doesn’t appear to be appropriate to measure participants’ “initial variability”. As pointed out by the authors in their response, the role of movement variability and outcome variability were differentiated in the existing literature. Thus, their measure should also be differentiated to provide clear findings.

Experimental design

I only have some minor comments regarding the new reported results:
l.268-270: The factor of the ANOVA to which these results refer to is not stated. The pairwise comparisons performed should be reported.
l.270: “Thus” should be removed (it is not mandatory that the interaction between the factors is also not significant).
l.275-277: This sentence can be removed. Pairwise comparisons should not be performed if the corresponding factor is not significant in the ANOVA.
l.283-284: The mentioned results should be reported. It would also be informative to report the means and corresponding variability values.
l.289-290: I still don’t understand what is the effect size g. I think that the results of the post hoc tests could be reported instead.
l.293: The authors previously reported that the group factor was not significant in the ANOVA (l.270). If so, this paragraph should be removed.
l.323-333: Correlations between VE and AE are confusing. Positive r values should mean that the higher the VE value in pretest (which seems to be the “initial VE”) the higher the increase in AE between pre-test and post-test. But as AE should be decreased, thus, this would mean that lower “initial VE” are related to higher improved. I suggest adding a sentence describing the relationship between “initial VE” and change in AE.

Validity of the findings

Most of my concerns regarding the findings presented in the discussion and conclusion were addressed. I still have concerns regarding the discussion of “initial variability” as “movement variability”, because variable error was measured.

Additional comments

As I remain concerned by the measure of "initial variability" used in this manuscript and by the corresponding conclusions, I will recommend once again a major revision.

---

## Round 0.4 · accepted · Accept

Thank you for your work to bring this manuscript up to an appropriate standard. Your responses to the reviewers suggestions have improved the manuscript and it is now clearly written and appropriate for publication.